# Phytochemical Screening and Biological Activities of *Lippia multiflora* Moldenke

**DOI:** 10.3390/molecules30132882

**Published:** 2025-07-07

**Authors:** Dorcas Tlhapi, Ntsoaki Malebo, Idah Tichaidza Manduna, Monizi Mawunu, Ramakwala Christinah Chokwe

**Affiliations:** 1Centre for Applied Food Sustainability and Biotechnology, Faculty of Health and Environmental Sciences, Central University of Technology, Bloemfontein 9300, South Africa; imanduna@cut.ac.za; 2Centre for Innovation in Learning and Teaching, Central University of Technology, Bloemfontein 9300, South Africa; nmalebo@cut.ac.za; 3Department of Agronomy, Polytechnic Institute, Kimpa Vita University, Uíge P.O. Box 77, Angola; m.mawunu2000@gmail.com; 4Department of Biology, Faculty of Science and Technology, University of Kinshasa, Kinshasa P.O. Box 190, Congo; 5Department of Chemistry, College of Science Engineering and Technology, University of South Africa, Johannesburg 1710, South Africa; chokwrc@unisa.ac.za

**Keywords:** *Lippia multiflora*, chemical profile, multivariate analysis, antioxidant activity, cytotoxicity

## Abstract

*Lippia multiflora* Moldenke is widely used in Angola, on the African continent, and beyond for the treatment of many health conditions such as hypertension, enteritis, colds, gastrointestinal disturbances, stomachaches, jaundice, coughs, fevers, nausea, bronchial inflammation, conjunctivitis, malaria, and venereal diseases. However, there is limited literature about the active compounds linked with the reported biological activities. This study aims to assess the chemical profiles, antioxidant properties, and the cytotoxicity effects of the roots, stem bark, and leaves of *L. multiflora*. Chemical characterization of the crude extracts was assessed through quantification of total phenolic and flavonoid contents followed by Q exactive plus orbitrap™ ultra-high-performance liquid chromatography-mass spectrometer (UHPLC-MS) screening. The correlation between the extracts and the correlation between the compounds were studied using the multivariate analysis. Principal component analysis (PCA) loading scores and principal component analysis (PCA) biplots and correlation plots were used to connect specific compounds with observed biological activities. The antioxidant activities of the crude extracts were carried out in vitro using DPPH (2,2-diphenyl-1-picrylhydrazyl) free radical scavenging and reducing power assays, while the in vitro toxicology of the crude extracts was evaluated using the 3-(4,5-dimethylthiazol-2-yl)-2,5-diphenyltetrazolium bromide (MTT) assay. A total of twenty constituents were characterized and identified using the UHPLC–Q/Orbitrap/MS. The methanol leaf extract showed the highest antioxidant activity in the DPPH free radical scavenging activity (IC_50_ = 0.559 ± 0.269 μg/mL); however, the stem bark extract had the highest reducing power (IC_0.5_ = 0.029 ± 0.026 μg/mL). High phenolic and flavonoid content was found in the dichloromethane leaf extract (32.100 ± 1.780 mg GAE/g) and stem bark extract (624.153 ± 29.442 mg QE/g), respectively. The results show the stem bark, methanol leaf, and dichloromethane leaf extracts were well-tolerated by the Vero cell line at concentrations up to 50 µg/mL. However, at 100 µg/mL onward, some toxicity was observed in the root, methanol leaf, and dichloromethane leaf extracts. The UHPLC–Q/Orbitrap/MS profiles showed the presence of terpenoids (n = 5), flavonoids (n = 5), phenols (n = 4), alkaloids (n = 3), coumarins (n = 1), fatty acids (n = 1), and organic acids (n = 1). According to several studies, these secondary metabolites have been reported and proven to be the most abundant for antioxidant potential. The identified flavonoids (catechin, quercitrin, and (−)-epigallocatechin) and phenolic compound (6-gingerol) can significantly contribute to the antioxidant properties of different plant parts of *L. multiflora*. The research findings obtained in this study provide a complete phytochemical profile of various parts of *L. multiflora* that are responsible for the antioxidant activity using UHPLC–Q/Orbitrap/MS analysis. Furthermore, the results obtained in this study contribute to the scientific information or data on the therapeutic properties of *Lippia multiflora* and provide a basis for further assessment of its potential as a natural remedy.

## 1. Introduction

*Lippia multiflora* Moldenke belongs to the Verbenaceae family, which includes about ~200 known species of small trees, shrubs, and grasses. It is usually found in tropical West African regions, Central American countries, and South American countries [1,2,3]. *Lippia multiflora* is commonly known as “Ti-tree, bush tea”, “Gambian tea bush”, or “healer herb” (in English), “thé de Gambie’’ (in French), “bunsurun fadama” or “godon kada” (in Hausa), “sre Numum” (in Akan), “naasuruu’’ (in Ga), “afudoti or afu” (in Ewe), and “Bulukutu” (in Kikongo) [4,5,6,7]. It is an aromatic, perennial plant with more or less woody stems growing up to 2.7 to 4.0 m in height [4]. Aerial and underground parts of *L. multiflora* are commonly used by traditional healers in Angola and other African countries for the treatment of health conditions such as hypertension, enteritis, cleaning lymph, chest pain, diabetes, colds, gastrointestinal disturbances, stomach aches, jaundice, coughs, fevers, nausea, bronchial inflammation, conjunctivitis, malaria, and venereal diseases [5,7,8,9].

Ethnopharmacological studies have shown antimalarial, antiviral, antifungal, analgesic, antipyretic, anti-inflammatory, antimicrobial, anti-diarrheal, and antifungal activities from *Lippia multiflora* [10,11,12,13]; these activities are usually attributed to the presence of alkaloids, terpenes, saponins, glycosides, flavonoids, and tannins [4]. Despite *L. multiflora* being used in traditional herbal medicine, the molecular basis underlying its therapeutic effects remains largely unexplored. Previous studies on the phytochemistry of *Lippia multiflora* have revealed very few isolated compounds, such as monoterpenoids (carvacrol and geniposide) and phenolic compounds (leucoseptoside A, alyssonoside, isoverba phenolscoside, verbascoside, samioside, isonuomiside A, and nuomioside A), using classical methods [14,15]. According to literature review studies [14,15], the pharmacological properties of these compounds isolated from *L. multiflora* have never been studied. Furthermore, the extracts and secondary metabolites of this plant species remain less explored with regard to their antioxidant and cytotoxic activities. Highlighting a need to identify, quantify, and characterize the metabolite profile of different organs of *Lippia multiflora* fully. Phytochemical screening allows for the identification of different classes of phytoconstituents within plant products, which could have potential pharmaceutical applications.

The current study shows that phytochemical analysis using UHPLC–Q/Orbitrap/MS enables a faster quantification, characterization, and identification of various secondary metabolites with remarkable sensitivity and accuracy compared to classical phytochemical techniques, which are time-consuming and sensitive, and consume a high amount of solvents. In addition, the quantity of chemical compounds isolated from plants is often small; thus, conducting structure elucidation of the isolated compounds from plants can be difficult to carry out using classical methods and requires precision. The distribution of antioxidants in different organs of the plant species has been evaluated using the DPPH (2,2-diphenyl-1-picrylhydrazyl) free radical scavenging and reducing power assays. Natural antioxidants in a daily diet have shown substantial benefits for human health and different diseases by reducing the effect of reactive oxygen or nitrogen compounds [16].

Oxidative stress is caused by the imbalance between the production of reactive oxygen species (ROS) and the body’s antioxidant defense mechanisms [17]. Free radicals that are released by cells often result in the degradation of proteins and DNA that facilitates diseases such as Alzheimer’s, cancer, neurodegenerative diseases, diabetes, age-related degeneration conditions, inflammation, rheumatoid arthritis, and atherosclerosis [18]. Antioxidants protect deoxyribonucleic acid (DNA) from oxidation to lower the risk of developing degenerative and chronic conditions [19]. Previous research studies have uncovered that herbal or traditional medicinal plants are good sources of antioxidants due to the fact that they are rich in flavonoids and phenolic compounds [20]. These compounds can protect humans from high levels of chelate metal ions, scavenge free radicals, and inhibit lipid peroxidation and free radicals [21,22]. A study conducted by Dabire et al. (2015) confirmed that the leaf crude extracts of *L. multiflora* have potent antioxidant activity against DPPH (2,2-diphenyl-1-picrylhydrazyl) free radical scavenging assays [23]. However, the antioxidant activity of the crude stem bark and root extract from *L. multiflora* has never been explored. Therefore, this study aimed at exploring the phytochemical and pharmacological properties of *L. multiflora*. It focuses on the crude leaf, root, and stem bark extracts, using UHPLC–Q/Orbitrap/MS, and reports on the antioxidant and cytotoxicity assessments.

## 2. Results and Discussion

### 2.1. Total Flavonoid Contents (TFCs) and Total Phenolic Contents (TPCs)

Natural antioxidants from plants are usually found in the form of phenolic compounds such as tocopherols, phenolic acids, flavonoids, etc. [24,25]. Several studies have shown that the antioxidant activity of medicinal plants could be attributed to their high content of flavonoids and phenolic compounds [26,27,28]. Phenolic compounds play a critical role in scavenging free radicals, reducing tocopherol radicals, activating antioxidant enzymes, chelating metals, and inhibiting enzymes that cause oxidation reactions, thus protecting cells from oxidative damage [21]. Flavonoids play a significant role in scavenging most oxidizing molecules, including various free radicals (peroxy radicals, hydroxyl radicals, and superoxide anion radicals) and singlet oxygen [22]. The different plant organs in *L. multiflora* were screened for the total contents of phenolics and flavonoids that might have contributed to their antioxidant activities. The total flavonoid content of the crude extracts was determined using the aluminum chloride (AlCl_3_) colorimetric method. Quercetin was used to formulate a standard curve (*y* = 0.0865*x* + 3.3762; R^2^ = 0.9973; Appendix A). The highest flavonoid content was found in methanol stem bark extract (624.153 ± 29.442 mg QE/g, *p <* 0.05), followed by methanol root extract (593.72 ± 57.918 mg QE/g, *p <* 0.05), dichloromethane leaf extract (305.666 ± 117.401 mg QE/g, *p <* 0.05), and methanol leaf extract (37.173 ± 6.443 mg QE/g, *p <* 0.05; see Figure 1). Our findings are in agreement with previous research studies that have shown that methanol stem bark extracts have high flavonoid content compared to the other different parts of the plant [29,30,31,32]. Moreover, other studies have proven that the antioxidant properties of several medicinal plant extracts are due to the high quality of the flavonoids or synergistic interactions of flavonoids with other various metabolites [33,34,35,36,37]. Therefore, our results confirm that the flavonoids found in the stem bark extracts of *L. multiflora* contributed to the potent antioxidant activities. Furthermore, the total phenolic content in the crude extracts was estimated using Folin–Ciocalteu’s reagent. Gallic acid was used to formulate a calibration curve (*y* = 1.0372*x* + 1.2487; R^2^ = 0.9076; see Appendix A). The results in Figure 1 show that total phenolic compounds in the extracts varied widely, ranging from 8.313 ± 3.179 to 32.100 ± 1.780 mg GAE/g with a descending order of dichloromethane leaf > root > methanol leaf > stem bark. The dichloromethane leaf extract exhibited the highest total phenolic content (32.100 ± 1.780 mg GAE/g, *p* < 0.05), while the stem bark extract (8.313 ± 3.179 mg GAE/g, *p* < 0.05) exhibited the lowest total phenolic content in comparison with the other crude extracts, as shown in Figure 1. The results obtained in this study support the result of a previous study conducted by Rouamba et al. (2024), who reported that the total phenolic content in the dichloromethane extract of *L. multiflora* leaves extracted by the maceration method was 96.70 ± 1.70 mg GAE/g (*p* < 0.05) [38]. The phenolic content of any plant is directly linked to its antioxidant properties. Phenolic compounds act as hydrogen donors and reducing agents and are capable of scavenging free radicals [39]. The presence of a noticeably good amount of phenolics in the leaf extracts of *L. multiflora* might contribute significantly to the antioxidant properties. These properties could be the reason why this plant species is used in various traditional herbal drugs. Furthermore, in other previous studies, the total phenolic contents of methanol and ethanol extracts of *L. multiflora* leaf was 462.20 ± 10.90 mg GAE/g (*p* < 0.05) and 71.17 ± 0.2 µg GAE/g [23,38], which is quite higher than that of the present study. Therefore, it can be suggested that the concentration of phenols that affect the antioxidant activities of the plant depends on the extraction methods used, the choice of extraction solvent, the metabolite content of each plant species, and other conditions [39,40]. Consequently, comparable with the data in the existing literature for other raw extracts of the plant products [41], our findings indicated that the contents of phenolics and flavonoids, as well as the reducing power and radical scavenging activities, might be due to the phenolic acids and flavonoids found in various extracts of *L. multiflora*.

### 2.2. Phytochemical Screening of the Crude Extracts of the Root, Leaf, and Stem Bark Using UHPLC–Q/Orbitrap/MS Analysis

The identification of the chemical constituents was carried out by UHPLC–Q/Orbitrap/MS. Both negative (ESI (−), Appendix A) and positive (ESI (+), Appendix A) modes were evaluated; however, the negative (ESI (−)) mode was chosen for additional sample investigation since it resulted in a larger abundance of ions. A total of twenty constituents from different plant parts of *L. multiflora* have been identified and characterized by comparing their spectral information with values in the existing literature. UHPLC–Q/Orbitrap/MS data for the identified chemical constituents, namely, the main product ions, the molecular ion [M − H]^−^, retention time, and their fragmentation ions, are provided in Table 1. Different classes of metabolites, such as terpenoids (**1**–**5**), flavonoids (**6**–**10**), phenols (**11**–**14**), alkaloids (**15**–**17**), coumarin (**18**), fatty acids (**19**), and organic acids (**20**), were identified. The UHPLC–Q/Orbitrap/MS profiles showed that various *Lippia multiflora* plant parts generated both similar and dissimilar compounds. Many of these compounds were mostly found in the leaf rather than in the stem and root extracts.

#### 2.2.1. Terpenoids

Compound **1** (Appendix A) with a precursor ion at *m*/*z* 483.12 [M − H]^−^ and the molecular formula C_30_H_44_O_5_ was identified as poricoic acid B; this molecule produced a product ion at *m*/*z* 483, which is similar to results previously reported in the literature [42]. Compound **2** (at *m*/*z* 455.30) exhibited a molecular ion peak that corresponded to boswellic acid alpha (Appendix A), which was verified using two fragment ions [M − H]^−^ at *m*/*z* 454 and 453 [43]. However, compounds **3** (Rt = 10.06 min) and **4** (Rt = 1.55 min), with [M − H]^−^ ions at *m*/*z* 469.11 (Appendix A) and *m*/*z* 501.11 (Appendix A), respectively, were identified as eburicoic acid (C_31_H_50_O_3_) and medicagenic acid (C_30_H_46_O_6_), respectively [44,45]. Compound **5** (Appendix A), with an [M − H]^−^ ion at *m*/*z* 525.30, which produced a fragmentation at *m*/*z* 525, was identified as albiflorin by comparison with the reference [46].

#### 2.2.2. Flavonoids

The MS-MS spectra of compounds **6** (Appendix A), **7** (Appendix A), **8** (Appendix A), **9** (Appendix A), and **10** (Appendix A) revealed fragment ions at *m*/*z* 259,231; *m*/*z* 259,243; *m*/*z* 267; *m*/*z* 311; and *m*/*z* 455,303, respectively. These data are similar to the published data on catechin (C_15_H_14_O_6_), epicatechin (C_15_H_14_O_6_), formononetin-7-*O*-glucoside (Ononin, C_22_H_22_O_9_), 3′,4′-dimethoxy-7-hydroxyflavone (C_17_H_14_O_5_), and quercitrin (C_21_H_20_O_11_), respectively [47,48,49,50].

#### 2.2.3. Phenolic Compounds

A total of four phenolic metabolites were identified based on the MS/MS fragment ions (*m*/*z*) observed, respective to the precursor or parent ions observed. The identification of these phenolic metabolites in the *Lippia multiflora* extract was executed by comparing the spectral data with data in the literature. Compounds **11** (*m*/*z* 293.18 [M − H]^−^), **12** (*m*/*z* 277.20 [M − H]^−^), **13** (*m*/*z* 295.23 [M − H]^−^), and **14** (*m*/*z* 305.92 [M − H]^−^) were identified as 6-gingerol (C_17_H_26_O_4_, Appendix A), 6-paradol (C_17_H_26_O_3_, Appendix A), esmolol (C_16_H_25_NO_4_, Appendix A), and (−)-epigallocatechin (C_15_H_14_O_7_, Appendix A), respectively [48,51]. The MS-MS spectra of compounds **11**, **12**, and **14** showed fragment ions at *m*/*z* 248, 237, and 209; *m*/*z* 283, 233, and 205; and *m*/*z* 289 and 221, respectively, whereas peak 13 did not show recognizable MS fragment ions.

#### 2.2.4. Alkaloids

A total of three alkaloid compounds were identified in various plant parts of *Lippia multiflora*. Compound **15** (Appendix A) was tentatively assigned as isomajdine (C_23_H_28_N_2_O_6_); fragmentation from [M − H]^−^ ions at *m*/*z* 397, 381, 369, 243, and 231 [52]. Based on the comparison of the MS data and retention times with those reported in the literature [53], compounds **16** (*m*/*z* 339.12 [M − H]^−^, Appendix A) and **17** (*m*/*z* 339.20 [M − H]^−^, Appendix A) were identified as yohimbinic acid (C_20_H_24_N_2_O_3_) and reserpic acid (C_22_H_28_N_2_O_5_).

#### 2.2.5. Coumarins and Others

Compound **18** (Appendix A) with a precursor ion at *m*/*z* 339.07 [M − H]^−^ and the molecular formula C_15_H_16_O_9_ was identified as 6,7-dihydroxycoumarin-6-glucoside (Esculin), a coumarin glucoside. This molecule produced a product ion at *m*/*z* 338 from the precursor ion of *m*/*z* 339.07 [54]. In addition to this, one fatty acid compound was identified with *m*/*z* 307.19 [M − H]^−^ as a precursor ion and confirmed as dihomolinoleic acid (compound **19**, Appendix A). Compound **19** generated fragment ions at *m*/*z* 289 and 290 [55]; the spectral data are similar to the values in the literature. Compound **20** (Rt = 0.35 min), with [M − H]^−^ ions at *m*/*z* 229.14 (Appendix A), was identified as dodecanedioic acid (C_12_H_22_O_4_), an organic acid. Compound **20** generated fragment ions at *m*/*z* 213; the data were confirmed by comparison with previous research [56].

### 2.3. Comparison of Analytical Equipment Used for Compound Identification in This Study with the Existing Literature

Table 2 summarizes differences in the analytical equipment used for compound identification in this study (exp) with the existing literature (lit). Desmiaty et al. (2020) used an UPLC-MS XEVO G2-XS QTOF to identify the presence of poricoic acid B (**1**) from the crude stem bark methanol extract of *Rubus fraxinifolius* Poir. [42]. Zhang et al. (2021) structurally characterized poricoic acid B (**1**) isolated from the crude ethyl acetate poriae cutis extract of *Poria cocos* (Schw.) Wolf using a UHPLC-QTOF-MS/MS and nuclear magnetic resonance (^1^H-NMR and ^13^C-NMR) spectroscopy [57]. Mannino et al. (2016) identified boswellic acid alpha (**2**) from the methanol gum resins of *Boswellia sacra* Flueck and *Boswellia serrata* Roxb by HPLC-DAD-ESI-MS/MS [43]. Compound (**3**) was tentatively characterized by UPLC-QTOF-MS from the ethanol cactus, cladodes, and callus of *Opuntia ficus-indica* (L.) Mill. [44]. In another study by Sheth et al. (1967), eburicoic acid (**3**) isolated from the petroleum ether stem decay extract of *Fomitopsis pinicola* (Sw.) P.Karst. was chemically characterized by means of FTIR and NMR spectroscopy [58]. Compound (**4**) was characterized as medicagenic acid from the root of *Medicago truncatula* Gaertn. using liquid chromatography coupled to negative-ion electrospray ionization Fourier transform ion cyclotron resonance mass spectrometry (LC ESI FT-ICR MS) [45]. However, Timbekova et al. (1996) identified and characterized medicagenic acid (**4**) from the leaves and root extracts of *Medicago sativa* L. by gas–liquid chromatography (GLC) and nuclear magnetic resonance (^1^H-NMR and ^13^C-NMR) spectroscopy [59]. Wang et al. (2016) detected the presence of albiflorin (**5**) in the crude ethanol extracts of aerial and underground plant parts of *Cinnamomi ramulus*, *Paeoniae radix* Alba, *Jujubae fructus*, *Glycyrrhizae radix* et *Rhizoma preparata* cum melle, and *Zingiberis rhizoma* Recens by LC-Q-TOF-MS and LC-IT-MS [46]. On the other hand, Wu et al. (2021) used HPLC–DAD and HPLC–DAD–ESI–MS to identify albiflorin (**5**) in the crude ethanol extracts (root, stem, leaves, and flowers) of *Paeonia lactiflora* Pall. [60]. Liu et al. (2021) used an UHPLC-Q-Orbitrap HRMS to identify catechin (**6**) in the traditional Chinese medicine decoction [47]. The herbal mixture contained *Pericarpium citri* Reticulatae, *Radix asteris*, *Radix scutellariae*, *Semen armeniaceae* Amarum, *Fructus aurantii* Immaturus, *Rhizoma belamcandae*, *Radix glycyrrhizae* Praeparata, *Agastachis herba*, *Rhizoma pinelliae*, *Gypsum fibrosum*, *Rhizoma dioscoreae*, *Poria cocos*, *Asari herba*, *Polyporus flos* Farfarae, *Alismatis rhizoma*, *Cinnamoni ramulus,* and *Ephedrae herba* macerated in deionized water before being decocted [47]. Furthermore, Abdel-Aal et al. (2022) separated and quantified catechin (**6**) using an UPLC in the aqueous tea leaves of *Camellia sinensis* (L.) Kuntze; the identity of catechin (**6**) was confirmed using liquid chromatography mass spectrometry (LC-MS/MS) [61]. Araujo et al. (2020) screened and identified epicatechin (7) from the aqueous leaves, pulp, and seed extracts of *Eugenia calycina* Cambess by LC-ESI-MS/MS [48]. Compound (**8**) was separated using an HPLC/DAD from the methanolic leaves and flower extracts of Sulla (*Hedysarum coronarium* L.); the chemical structure of formononetin-7-*O*-glucoside (Ononin, **8**) was identified by LC-ESI-MS [49]. Conversely, Zgórka et al. (2022) identified and quantified formononetin-7-*O*-glucoside (Ononin, **8**) from the aqueous–ethanolic leaves and flower extracts of Zigzag clover (*Trifolium medium* L.) using advanced analytical techniques such as *RP-LC/PDA/ESI-QTOF/MS-MS* spectroscopy [62]. Tine et al. (2017) used LC-MS/MS to characterize quercitrin (**10**) from the methanolic plant organs from separated parts (root barks, trunk barks, stem, leaves, and fruits) of *Zanthoxylum zanthoxyloides* (Lam.) Zepern. & Timler [50]. In another study, Parolin Trindade et al. (2025) characterized quercitrin (10) from the crude ethanolic leaf extract of *Lippia origanoides* Kunth by UHPLC-ESI-QTOF-MS/MS [63]. Araujo et al. (2020) screened and identified 6-gingerol (**11**), 6-paradol (**12**), and esmolol (**13**) from the aqueous leaves, pulp, and seed extracts of *Eugenia calycina* Cambess by LC-ESI-MS/MS [48]. Compound (**14**) was tentatively identified as (−)-epigallocatechin from a Japanese knotweed rhizome bark extract by HPLC-MS, SEC-HPLC-UV, and HPTLC [64]. Gülçin et al. (2012) isolated and characterized isomajdine (**15**) from the crude methanol root extract of *Vinca herbacea* Waldst. and Kit by NMR [65]. In addition, Kumar et al. (2016) used HPLC–ESI–QTOF–MS/MS to determine the chemical structure of yohimbic acid (**16**) and reserpic acid (**17**) from the crude ethanol root extracts of six *Rauwolfia* species (*R. vomitoria*, *R. verticillata*, *R. tetraphylla, R. serpentina, R. micrantha,* and *R. hookeri*) [53]. In another study conducted by Wang et al. (2014), yohimbic acid (**16**) was identified from *Pausinystalia johimbe* (K. Schum.) Pierre ex Beille by UHPLC–UV–MS/MS [66]. Compound (**18**) was tentatively characterized by HR-LCMS from the crude methanol extracts (leaves, young stems, and roots) of *Oroxylum indicum* (L.) Vent [54]. The chemical structures of eicosadieneoic acid (**19**) and dodecanedioic acid (**20**) were determined by LC-Q-TOF-MS and UHPLC–Q-exactive orbitrap MS from the methanol and ethyl acetate seed extracts of *Nigella sativa* L. and the water rhizome extracts of *Senecio cannabifolius* Less., respectively [55,56].

### 2.4. Multivariate Analysis

The phytochemical data were analyzed using the PAST 4.03 software to assess the variance and correlation between the extracts and the correlation between the compounds. The compounds were given code names (**C1**–**C20**) according to their order in Table 1. The variance and correlation between the extracts were assessed using principal component analysis and hierarchical clustering. In contrast, the correlation between the compounds that were tentatively identified was evaluated using the correlation plot. Principal component 1 (PC1) and principal component 2 (PC2) explain 52.23% and 30.89% of the variance, respectively (Figure 2A). The most influential compounds along PC1 were **C9**, **C10**, **C11**, **C13**, **C16**, **C18**, and **C4**, **C5**, and **C19**, which had a positive and negative influence, respectively. While **C6**, **C7**, and **C15** had a positive influence, and **C3** and **C19** had a negative influence on PC2, as shown in Table 3. This means that variance along PC1 and PC2 is due to the presence or absence of these compounds in the extracts. The PCA biplot shows that DL and ML are positively correlated to each other but are negatively correlated to R and S. This is because **C10** and **C18** were tentatively identified in these extracts while they were not detected in R and S. Figure 2B shows that DL and ML are more similar to each other than they are to other extracts. This is due to the fact that **C10** and **C18** were identified in these extracts, while they were not identified in the other extracts. At the same time, the same is true for R and S. Figure 2C, however, shows that R and S correlate more to DL than to ML. This is because ML contains **C11**, **C13**, and **C16**, while these compounds were not identified in the other extracts. Figure 2D shows that **C9**, **C11**, **C13**, and **C16** are highly correlated; this is because these compounds were all identified in R, S, and DL. **C18** and **C10** are correlated because they were identified in DL and ML but not in R and S. **C5** and **C4** were identified in S, while not in the other extracts. **C7** and **C6** were identified in S and DL but not in R and ML; hence, they have a high correlation. **C10** and **C18** are negatively correlated to **C19**; this is due to the fact that **C10** and **C18** were identified in ML and DL, while **C19** was not. Furthermore, **C19** was identified in R and S, while **C18** and **C10** were not identified in these extracts.

### 2.5. Antioxidant Activity

All samples exhibited strong free radical scavenging and reducing power activities (Table 4). The methanol leaf extract exhibited high radical scavenging activity with an IC_50_ value of 0.559 ± 0.269 µg/mL (*p* < 0.05), whereas the root extract exhibited the lowest radical scavenging activity (IC_50_ = 25.108 ± 19.798 µg/mL, *p* < 0.05). In contrast, the stem bark extract exhibited high reducing power activity with an IC_0.5_ value of 0.029 ± 0.026 µg/mL (*p* > 0.05), while the methanol leaf extract exhibited the lowest reducing power activity (0.710 ± 1.220 µg/mL (*p* > 0.05)), as shown in Table 4. The presence of the polyphenols and flavonoids identified in the methanol leaf and stem bark extracts could have contributed to the high antioxidant properties of these extracts, as shown in Table 4. The reason for the high radical scavenging activity of the methanol leaf extract as compared to the other extracts could be that it contains **C9**, **C11**, **C13**, and **C16**, whose antioxidant properties have been reported before, while the other extracts do not. This is also shown by Table 3 and Figure 2A, where it is observed that these compounds have a positive influence on PC1, meaning extracts with a high PC1 score contain these compounds. The compounds **C4** and **C5**, which have also shown antioxidant activity before, were only identified in the stem bark extract, which could be the reason why this extract exhibited high reducing power activity compared to the others. These compounds have a negative influence on PC1; this means extracts with a low PC1 score contain these compounds. Additionally, the correlation of these compounds is also shown by Figure 2D. Wang et al. (2003) reported that 6-gingerol (**11**) isolated from the rhizome of *Zingiber officinale* Roscoe exhibited a high DPPH radical scavenging activity, with an EC_50_ value of 23.07 µg/mL (*p* < 0.05) [67]. However, Jug et al. (2021) showed that (−)-epigallocatechin (**14**), tentatively identified from a Japanese knotweed rhizome bark extract, exhibited high DPPH radical scavenging activity, with an IC_50_ value of 1.5 µg/mL^−1^ [64]. Grzesik et al. (2018) reported that catechin (**6**) found in the stem bark extract (Table 1) has potent antioxidant activities against DPPH radical scavenging and reducing power activities, with IC_50_ and IC_0.5_ values ranging from 0.965 ± 0.015 mol TE/mol to 3.965 ± 0.067 mol TE/mol (*p* < *0.05*) and 0.793 ± 0.004 mol TE/mol to 1.032 ± 0.007 mol TE/mol (*p* < *0.05*), respectively [68]. Furthermore, Picos-Salas et al. (2024) confirmed that the presence of quercitrin (**10**), quantified using an ultra-performance liquid chromatography mass spectrometry (UPLC-MS) analysis, contributed to the antioxidant activity of the methanolic stems, leaves, and flowers of *Lippia graveolens* Kunth using the oxygen radical absorbance capacity (ORAC) and trolox equivalent antioxidant capacity (TEAC) assays [69]. However, González Güereca et al. (2007) reported that quercitrin (**10**) isolated from the ethyl acetate stem of *Lippia graveolens* HBK var. *berlandieri* Schauer by silica gel column chromatography showed a high DPPH radical scavenging activity, with an IC_50_ value of 11.24 ± 1.10 µg/mL [70]. With that being said, it can be concluded that 6-gingerol (**11**), (−)-epigallocatechin (**14**), catechin (**6**), and quercitrin (**10**) play a significant role in the antioxidant activities of the methanol leaf and stem bark extracts. In addition, the findings obtained in this study are not in line with previous reports conducted by Dabire et al. (2015), who demonstrated that the dichloromethane leaf extracts of *L. multiflora* (IC_50_ = 175.60 ± 0.1 µg/mL) extracted by cold maceration had the lowest antioxidant activity using the 2,2-diphenyl-1-picrylhydrazine (DPPH) scavenging assay [23]. Dabire et al. (2015) further showed that the ethyl acetate leaf extracts of *L. multiflora* had the highest antioxidant activity, with an IC_50_ value of 23.68 ± 0.2 µg/mL [23]. These observations prove that the activity of a plant-extracted product depends not only on the existence of bioactive constituents but also on the possible interactions with other compounds present in the extract, the solvent used for extraction, and environmental factors (such as humidity, different soil characteristics, light intensity, temperature, rainfall, height above MSL, and change of season) [71,72,73]. The findings of this research also showed that the crude extracts had higher antioxidant activities compared to quercetin, gallic acid, and ascorbic acid using both 2,2-diphenyl-1-picrylhydrazine (DPPH) scavenging and reducing power methods. This shows that *L. multiflora* could be a great antioxidant agent useful in the treatment and management of free radicals.

### 2.6. Comparison of the Methods Used to Determine the Antioxidant Activity of Extracts Obtained in This Study with the Existing Literature

Table 5 summarizes differences in the methods used to determine the antioxidant activity of extracts obtained in this study (exp) with the existing literature (lit). Desmiaty et al. (2020) confirmed that the presence of poricoic acid B (**1**), identified using a liquid chromatography equipped with mass spectrometry (Waters UPLC–MS XEVO G2–XS QTOF, Milford, MA, USA) analysis, contributed to the antioxidant activity of the crude methanol stem bark extracts of *Rubus fraxinifolius* Poir. (IC_50_ = 63.04 ppm) using DPPH (2,2-diphenyl-1-picrylhydrazyl) free radical assays [42]. Gupta et al. (2022) reported that boswellic acid (**2**) alpha, characterized using high-performance liquid chromatography (HPLC), contributed to the antioxidant activity of the methanol oleo-gum resin extract of *Boswellia serrata* Roxb. with percent inhibition (70.7–78.4%) and (67.79–87.57%) at concentrations of 100 μg/mL and 500 μg/mL using DPPH (2,2-diphenyl-1-picrylhydrazyl) free radical scavenging and ABTS (2,2′-azino-bis(3-ethylbenzothiazoline- 6 sulfonic acid)) assays [74]. Wu et al. (2021) showed that albiflorin (**5**) found in the ethanol extracts (root, stem, leaves, and flowers) of *Paeonia lactiflora* Pall. could be responsible for the antioxidant effectiveness of this plant species against DPPH (IC_50_ = 55.05 ± 0.15 μg/mL) free radical scavenging and ABTS (IC_50_ = 33.10 ± 0.41 μg/mL) assays [60]. A study conducted by Grzesik et al. (2018) showed that catechin (**6**) and epicatechin (**7**) had the highest ABTS scavenging capacity and ferric reducing antioxidant power (FRAP), with IC_50_ values of 3.965 ± 0.067 mol TE/mol and 0.793 ± 0.004 mol TE/mol and 2.800 ± 0.051 mol TE/mol and 0.917 ± 0.004 mol TE/mol, compared to other metabolites [68]. Zgórka et al. (2022) documented that formononetin-7-*O*-glucoside (Ononin, **8**) obtained in the aqueous–ethanolic leaves and flower extracts of Zigzag clover (*Trifolium medium* L.) had higher antioxidant activity against ABTS radical (IC_50_ = 30.30 ± 0.18 µg/mL) and DPPH radical (IC_50_ = 30.18 ± 0.37 µg/mL) [62]. Tine et al. (2017) showed that the presence of quercitrin (**10**) was responsible for the antioxidant activities of the methanolic trunk barks (IC_50_ = 33.32 ± 0.88 µg/mL) and leaves (IC_50_ = 24.96 ± 0.32 µg/mL) of *Zanthoxylum zanthoxyloides* (Lam.) Zepern. & Timler using the ABTS radical method [50,69,70]. Furthermore, Wang et al. (2003) revealed that 6-gingerol (**11**) isolated from the rhizome of *Zingiber officinale* Roscoe displayed a high DPPH radical scavenging activity, with an EC_50_ value of 23.07 µg/mL [67]. Jug et al. (2021) showed that (−)-epigallocatechin (**14**), tentatively identified from a Japanese knotweed rhizome bark extract, exhibited high DPPH radical scavenging activity with an IC_50_ value of 1.5 µg/mL^−1^ [64]. Gülçin et al. (2012) reported that isomajdine (**15**) isolated and characterized from the crude methanol root extract of *Vinca herbacea* Waldst. and Kit exhibited antioxidant activity against ABTS scavenging (EC_50_ = 34.11 μg/mL), DMPD scavenging (EC_50_ = 32.09 μg/mL), and ferrous ion (Fe^2+^) chelating activity (EC_50_ = 10.05 μg/mL) [65].

### 2.7. Pearson Correlation Analysis Between Phytochemicals (TFC and TPC) and Their Antioxidant Activities

As shown in Table 6, a Pearson correlation was conducted between phytochemicals (TFC and TPC) and their antioxidant capacities to understand their relationship. The results illustrate a significant strong negative correlation from the reducing power with TFC (r = −0.956). DPPH activity showed a strong positive correlation with the TFC value (r = 0.619). The weak negative correlation (r = −0.019) between RP and TPC indicates no relation between reducing power and the total phenolic content. Additionally, the weak negative correlation (r = −0.111) between TPC and TFC indicates no relation between the total phenolic content and total flavonoid content. There was a moderate negative correlation (r = −0.406) between RP and DPPH activity. On the other hand, DPPH activity showed a weak relationship (r = 0.071) with TPC.

### 2.8. Cytotoxicity

Cytotoxicity was evaluated using the 3-(4,5-dimethylthiazol-2-yl)-2,5-diphenyltetrazolium bromide (MTT) assay against Vero cells (derived from the kidney of an African green monkey) at varying concentrations of 50 µg/mL, 100 µg/mL, 250 µg/mL, and 500 µg/mL (Figure 3). The inhibitory concentration value (IC_50_, Table 7) was evaluated from the percentage (%) of cell viability compared to the positive control (doxorubicin, Table 7 and Figure 3). The results show the stem bark, methanol leaf, and dichloromethane leaf extracts were well-tolerated by the Vero cell line at concentrations up to 50 µg/mL. On the other hand, at 100 µg/mL onward, some toxicity was observed in the root, methanol leaf, and dichloromethane leaf extracts, except in the stem bark (Figure 3). This might be due to the fact that the stem bark contains medicagenic acid (**4**) and albiflorin (**5**) which the other extracts do not. The only difference between the stem bark extract and the other extracts is also shown in the PCA loading scores (Table 3), where **C4** and **C5** have a negative influence on PC1, meaning samples with high PC scores do not contain these compounds. Furthermore, certain compounds present in the root, stem bark, and leaf extracts of *Lippia multiflora* (Table 1), such as eburicoic acid (**3**) [75], catechin (**6**) [76], epicatechin (**7**) [76], 6-gingerol (**11**) [77], (−)-epigallocatechin (**14**) [78], isomajdine (**15**) [65], and boswellic acid alpha (**2**) [79], are known to have various beneficial activities on the cytotoxicity of African green monkey (Vero) kidney cells. Therefore, these compounds may therefore be partly responsible for the observed activity. According to the literature, the toxicity of *Lippia multiflora* has not been determined using Vero cells.

## 3. Materials and Methods

### 3.1. General Experimental Procedure

All solvents and chemicals used in this study were of analytical grade. African green monkey (Vero) kidney cells (ATCC CCL-81) were purchased from Cytiva Hyclone, Marlborough, MA, USA. Minimal essential medium (MEM), Earle’s balanced salt solution (EBSS), L-glutamine, fetal bovine serum (FBS), penicillin/streptomycin (P/S), trypsin/EDTA, and trypan blue were from (Cytiva Hyclone, Marlborough, MA, USA). Dichloromethane, methanol, formic acid, and acetonitrile were purchased from Sigma-Aldrich (St. Louis, MO, USA). Dimethyl sulfoxide (DMSO), ascorbic acid, quercetin, gallic acid, 2,2-diphenyl-1-picrylhydrazyl (DPPH), potassium hexacyanoferrate [K_3_Fe (CN)_6_], trichloroacetic acid, ferric chloride, Folin–Ciocalteu, sodium carbonate (Na_2_CO_3_), aluminum chloride, and thiazolyl blue tetrazolium bromide were purchased from Sigma-Aldrich (Darmstadt, Germany). Doxorubicin was purchased from Merck (Darmstadt, Germany).

### 3.2. Sampling and Extraction

The various plant parts (root, stem bark, and leaf) of *Lippia multiflora* were collected at Uíge Province, Angola. Identification of the species was carried out by Mr. Monizi Mawunu of the Department of Agronomy, Polytechnic Institute, Kimpa Vita University, Angola. A voucher herbarium specimen, NM 01, was assigned, and the voucher was deposited in the herbarium of the Department of Agronomy. The samples were air-dried for four weeks and then ground to a fine powder using a POLYMIX^®^ lab mill with blade grinding, PX-MFC 90 D (Westdene, Bloemfontein, South Africa). Approximately ~220.22 g ground root and ~213.55 g stem bark of *Lippia multiflora* were each macerated in methanol (2 L) for two days, respectively. The stem bark and root extracts were filtered and then concentrated at 45 °C using a Büchi^®^ Rotavapor^®^ (Sigma-Aldrich, St. Louis, MO, USA) to obtain 4.19 g and 2.65 g of solid crude extracts. Moreover, ~217.58 g ground leaves of *Lippia multiflora* were macerated in dichloromethane (2 L) and methanol (2 L) for two days. The leaf extracts were filtered and then concentrated at 45 °C using a Büchi^®^ Rotavapor^®^ (Sigma-Aldrich, St. Louis, MO, USA) to obtain 3.53 g dichloromethane extract and 6.56 g methanol extract.

### 3.3. Total Flavonoid Content (TFC)

The total flavonoid content (TFC) of the extracts was determined by the aluminum chloride (AlCl_3_) method in Tlhapi (2018) [80]. The crude extracts (100 µL) were added in triplicate in 96-well plates containing 80 µL of distilled water, followed by an addition of aluminum chloride (2%, 100 µL) solution. The mixture was left to stand for 30 min, and the absorbance was read at 420 nm using a SoftMax^R^ Pro 6 (version 6.3) microplate reader. Quercetin was used as a standard to convert the absorbance of the crude extract to total flavonoid content expressed in milligram quercetin equivalent per gram of the crude extract (mg QE/g). The total flavonoid content of the crude extracts was determined using the aluminum chloride (AlCl_3_) colorimetric method. Quercetin was used to formulate a standard curve (*y* = 0.0865*x* + 3.3762; R^2^ = 0.9973; Appendix A).

The extracts were calculated using the following equation:C = c × V/m(1)
C = total amount of compounds.c = concentration of standard from the standard curve (mg/mL).V = volume of the extract (mL).m = weight of extract (g).

### 3.4. Total Phenolic Content (TPC)

The total polyphenolic contents (TPCs) of the extracts were determined by the Folin–Ciocalteu method as seen in Tlhapi (2018) [80]. The crude extracts (100 µL) were added in triplicate to a 96-well plate containing 80 µL of distilled water. A Folin–Ciocalteu reagent (10%; 20 µL) and 60 µL of 7% sodium carbonate (Na_2_CO_3_) solution were added to the mixture, respectively. The absorbance was read at 760 nm using a SoftMax^R^ Pro 6 (version 6.3) microplate reader. The gallic acid standard curve was used to convert the absorbance of the crude extracts to total phenolic content expressed in milligram gallic acid equivalent per gram of the crude extract (mg GAE/G). The total phenolic content in crude extracts was estimated using Folin–Ciocalteu’s reagent. Gallic acid was used to formulate a calibration curve (*y* = 1.0372*x* + 1.2487; R^2^ = 0.9076; see Appendix A). The extracts were calculated using Equation (1).

### 3.5. UHPLC-Q Exactive-Orbitrap-MS Analysis

The crude extracts were identified using ultra-high-performance liquid chromatography coupled with quadrupole-orbitrap MS (UHPLC-Q-orbitrap MS, Thermo Fisher Scientific, Waltham, MA, USA). LC-ESI-Orbitrap-MS analysis was executed using a 4.6 mm × 150 mm, particle size 3.5 μm C18 analytical column to separate the extracts. Mobile phases were formic acid (0.1 percent *v*/*v*) in water (solvent A) and formic acid (0.1 percent *v*/*v*) in acetonitrile (solvent B). The linear gradient elution began with five percent of solvent B and increased to a hundred percent of solvent B for twenty minutes. A sample injection volume of ten microliters, a mobile phase flow rate of 0.9 mL min^−1^, and a column temperature of 25 °C were used. Data processing was determined using the XCaliber (Thermo Fisher Scientific Inc., Waltham, MA, USA, version 3.0 software). Both negative (ESI (−), Appendix A) and positive (ESI (+), Appendix A) modes were evaluated; however, the negative (ESI (−)) mode was chosen for additional sample analysis since it resulted in a larger abundance of ions. Moreover, the mass spectrometry raw data files were converted to an open-source format (.mzXML) using ProteoWizard tool MSConvertGUI (version 3.0.24164-38d6037) software. The mzXML files were evaluated using MZmine version 3.9.0 software; centroid techniques such as mass detection, chromatograph building, and peak deconvolution were used to process settings of MZmine version 3.9.0. The aligned peak list, which contained the retention time, *m*/*z* values, and peak heights in each sample, was exported in CSV format. The exported file was uploaded to MassBank, an online database for compound identification. Phytochemicals with a match of ≥80% in the database were considered; the identified compounds were extracted from the sample chromatogram data for further validation.

### 3.6. Antioxidant Activity Assays

#### 3.6.1. Free Radical Scavenging Assay (DPPH)

The DPPH free radical scavenging ability was evaluated for the crude extracts using the DPPH (2,2-diphenyl-1-picrylhydrazyl) assay using a modified spectrophotometric method of Motamed and Naghibi (2010) [81]. The crude extracts (100 µL) were added to a 96-well plate containing distilled water (100 µL). After serial dilution, a 0.3 M DPPH/methanol (200 µL) solution was added to the mixture. The mixture was left at room temperature for 30 min, and the absorbance was measured using a SoftMax^R^ Pro 6 (version 6.3) microplate reader at 517 nm. All the crude extracts were tested in triplicate.

The percentage of radical scavenging was calculated by the following formula:% Free RSA = [(A_DPPH_ − A_sample_)/(A_DPPH_)] × 100(2)
where A_DPPH_ is the absorbance of the DPPH solution and A_sample_ is the absorbance of the samples and DPPH. The ability of the crude extracts to inhibit 50% of the free radical (IC_50_) was extrapolated from a graph of % RSA against concentration.

#### 3.6.2. Reducing Power

The reducing capacity of the crude extracts was estimated as described by Pereira et al. (2013) [82]. Fifty microliters of the crude extracts were mixed with fifty microliters of sodium phosphate buffer (0.2 M, pH 6.6). After serial dilution, a potassium hexacyanoferrate (1% aqueous, 50 µL) solution was added. A trichloroacetic acid (10%, 50 µL) solution was added after 20 min of incubation at 50 °C. Eighty microliters of each mixture was moved to another 96-well plate containing eighty microliters of distilled water. Ferric chloride (0.1%, *w*/*v*; 16 µL) was added, and the absorbance was measured using a SoftMax^R^ Pro 6 (version 6.3) microplate reader at 700 nm. All the crude extracts were tested in triplicate.

### 3.7. Cytotoxicity Assay

#### 3.7.1. Cell Culture

African green monkey (Vero) kidney cells (ATCC CCL-81, Cellonex, Johannesburg, South Africa) were cultured in Minimal Essential Medium (MEM), which consisted of Earle’s Balanced Salt Solution (EBSS) and L-Glutamine (2.0 mM) (Cytiva Hyclone, Marlborough, MA, USA) and was supplemented with 10% fetal bovine serum (FBS) and 1% penicillin/streptomycin (P/S, Biowest, Nuaille, France). These cells were incubated for 48 h at 37 °C in a 95% air and 5% CO_2_ humidified incubator (Nϋve, Ankara, Turkey). The cells were trypsinized with 0.25% trypsin/EDTA (Cytiva Hyclone, Marlborough, MA, USA) and split at a ratio of 1:5 for further passaging at 70 to 80% confluency. Trypan blue 0.4% (Cytiva Hyclone, Marlborough, MA, USA) on an automated cell counter (NanoEntek, Guro-gu, Seoul, Republic of Korea) was used to check the cell viability, and only suspensions with a cell viability higher than 90% were used for the cytotoxicity assay.

#### 3.7.2. Cytotoxicity Assay

The 3-(4,5-dimethylthiazol-2-yl)-2,5-diphenyltetrazolium bromide (MTT) cytotoxicity assay was determined as described by Mariri et al. (2024) [83]. The cytotoxic effects of the crude extracts of *Lippia multiflora* were evaluated against Vero cells. The Vero cells were seeded at a density of ten thousand cells on the 96-well microliter plates, and the cells were allowed to attach overnight. After incubation, the cells were treated with varying concentrations of 50 µg/mL, 100 µg/mL, 250 µg/mL, and 500 µg/mL dissolved in dimethyl sulfoxide and further diluted in fresh culture medium. In each test, the concentration of dimethyl sulfoxide in the medium did not surpass 0.5% (negative control). Doxorubicin was used as a positive control at varying concentrations of 0.67 µg/mL, 1.35 µg/mL, 2.71 µg/mL, and 5.43 µg/mL. The plates were further incubated for two days; thereafter, the culture medium containing tested samples was discarded and replaced by two hundred microliters of fresh medium with thirty microliters of thiazolyl blue tetrazolium bromide (5 mg/mL) dissolved in phosphate buffered saline (PBS). After 4 h of incubation, the culture medium was gently aspirated, and the formazan crystals were dissolved in fifty microliters of dimethyl sulfoxide. The mixture was kept in the dark for 15 min, and the absorbance was measured spectrophotometrically at 570 nm on a SpectraMax iD3 multi-mode microplate reader (Winooski, VT, USA). The cell viability was calculated as the percentage (%) of negative control (0.5% dimethyl sulfoxide; see Equation (3)).

The extracts were calculated using the following equation:(3)$$Cell\;viability\left(\%\right)=\frac{Abs\left(sample\right)-Abs\left(Blank\right)}{Abs\left(negative\;control\right)-Abs\left(Blank\right)}\times 100$$
where Abs is the absorbance.

### 3.8. Statistical Analysis

Statistical analysis was carried out using the IBM SPSS Statistics package, version 22 (Chicago, IL, USA). For the antioxidant, total flavonoid content, total phenolic content, and cytotoxicity tests, the measured data were obtained in triplicate, and the data were expressed as mean ± standard deviation. Mean differences of the crude extracts were assessed by a one-way analysis of variance (ANOVA) with Duncan’s Multiple Range Test (DMRT) in the antioxidant, total flavonoid content, and total phenolic content tests; the difference with *p* < 0.05 was considered statistically significant. A principal component analysis (PCA) biplot was prepared with both Y and X scaling at 10 to −10 and 95% ellipses; the eigenvalue scale was also used. Hierarchical clustering (prepared with the X scaling at 0 to 5, Y scaling at 0 to −0.3223, and 95% ellipses), classical clustering plot, neighbor-joining cluster plot, and the correlation plot were performed using the PAST 4.03 software. A Pearson correlation analysis was conducted using the IBM SPSS Statistics package, version 22 (Chicago, IL, USA), to evaluate the relationship between phytochemicals (TFC and TPC) and their antioxidant activities. All the extracts were analyzed in triplicate for the PCA and correlation plot methods. For cytotoxicity, the data were shown as the Vero cell number ± standard deviation at varying concentrations. Doxorubicin was used as a positive control drug standard for the cytotoxicity assay.

## 4. Conclusions

In this study, the phytochemical composition, total flavonoid content, total phenolic content, and the antioxidant and cytotoxic potentials of different organs of *Lippia multiflora* were evaluated. UHPLC–Q/Orbitrap/MS was used to detect 20 metabolites. The metabolite profiles showed that different plant parts of *L. multiflora* produce similar and dissimilar metabolites. Furthermore, the multivariate analysis of the data obtained showed that the chemistry of the root extracts and stem bark extracts is similar and different from the chemistry of the methanol leaf extracts and dichloromethane leaf extracts. The study demonstrated that the leaf and stem bark crude extracts had a greater antioxidant capacity compared to the root extract. The study also demonstrated that the root extract was more toxic at varying concentrations compared to the stem bark and leaf extracts. Furthermore, the efficient antioxidant activities were supported by the presence of a good amount of total flavonoid and phenolic content in the extracts. Our results highlight that this species provides a promising source for obtaining phytoconstituents that might be used to develop drugs that can be used in the treatment of infectious diseases. Bioassay-guided fractionation and isolation of phytochemicals are required to discover other bioactive compounds contributing to the biological activities of *L. multiflora*. Finally, in vitro and in vivo toxicological and pharmacological studies using different in vitro and in vivo cytotoxicity assays must be performed to confirm the safety of the identified compounds before considering *Lippia multiflora* as a viable candidate for drug development. Nonetheless, our results may represent a promising milestone in the medical or pharmaceutical industry’s search for new and effective therapies.

## Figures and Tables

**Figure 1 molecules-30-02882-f001:**
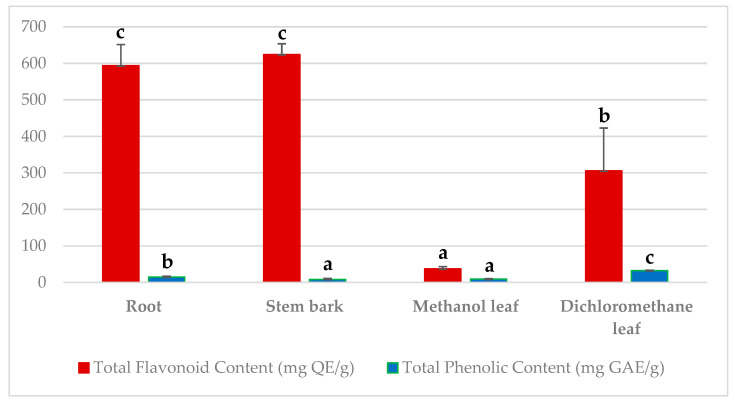
Total flavonoid content and total phenolic content of crude extracts. A different superscript letter indicates significant differences using a one-way ANOVA at *p <* 0.05. Data (n = 3) expressed as mean ± standard deviation. In the total flavonoid content test: ^a^—methanol leaf extract was significantly different from the stem bark extract, root extract, and dichloromethane leaf extract; ^b^—dichloromethane leaf extract was significantly different to all the crude extracts; ^c^—stem bark extract was not significantly different from the root extract. In the total phenolic content test: ^a^—methanol leaf extract was not significantly different from the stem bark extract; ^b^—root extract was significantly different from the stem bark extract, methanol leaf extract, and dichloromethane leaf extract; ^c^—dichloromethane leaf extract was significantly different from all the crude extracts.

**Figure 2 molecules-30-02882-f002:**
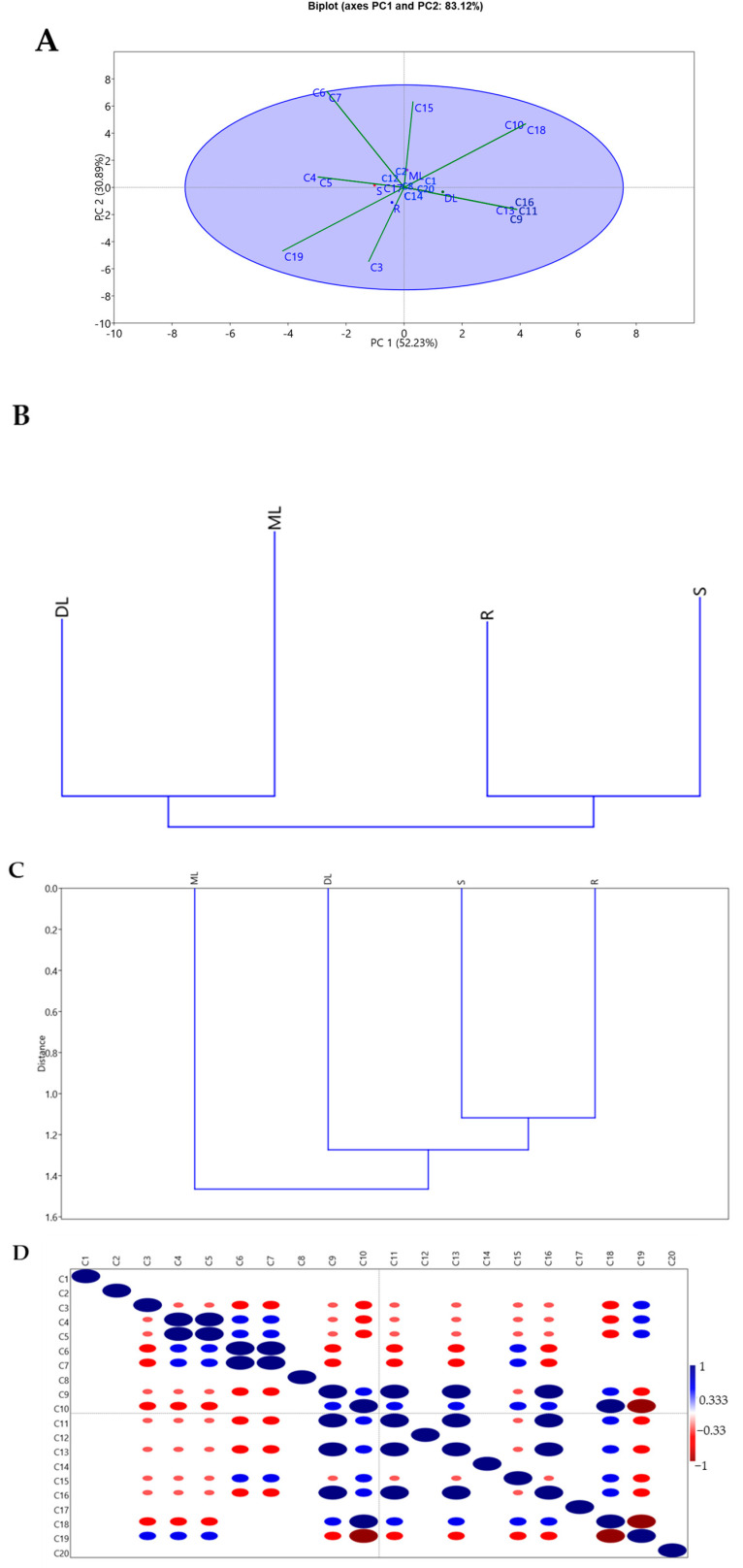
(**A**) PCA biplot of the extracts and the compounds that were tentatively identified. (**B**) The classical clustering plot of the extracts. (**C**) Neighbor-joining cluster plot of the extract. (**D**) Correlation plot of the compounds that were tentatively identified in the extracts.

**Figure 3 molecules-30-02882-f003:**
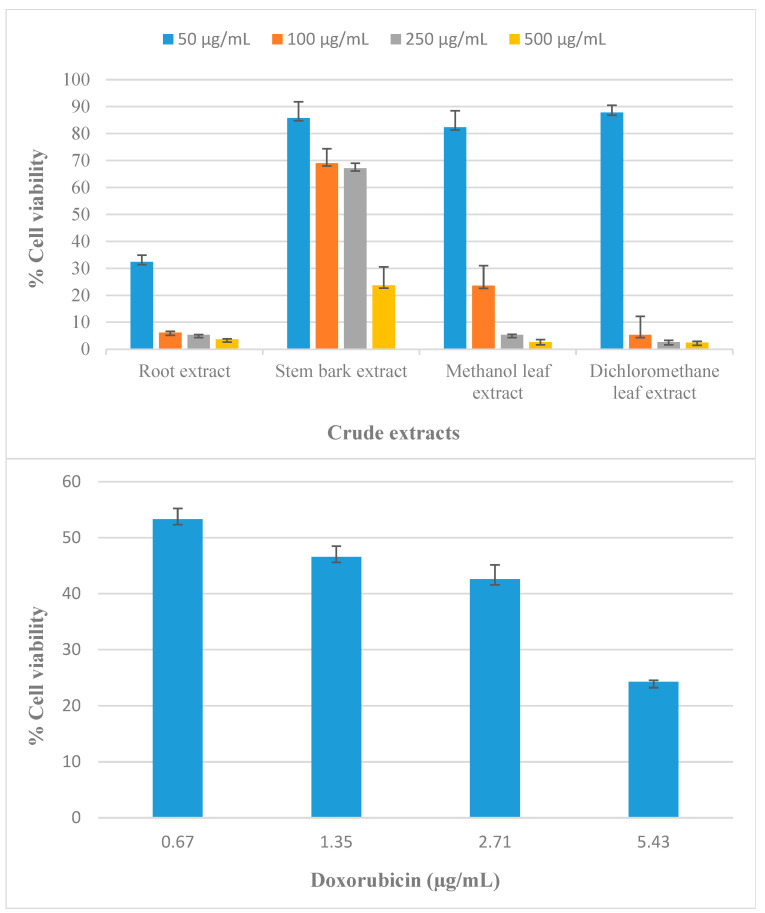
Cytotoxicity of *Lippia multiflora* extracts against Vero cells at varying concentrations of 50 µg/mL, 100 µg/mL, 250 µg/mL, and 500 µg/mL. Mean differences of samples were assessed by a one-way analysis of variance (ANOVA) with Duncan’s Multiple Range Test (DMRT); the difference with *p* < 0.05 was considered statistically significant. Error bars indicate the standard deviation of triplicate values conducted as a single experiment. All experiments were carried out in triplicate (n = 3), and data represent the mean ± standard deviation of two independent experiments. Data for doxorubicin (positive control) at varying concentrations of 0.67 µg/mL, 1.35 µg/mL, 2.71 µg/mL, and 5.43 µg/mL are shown for comparison.

**Table 1 molecules-30-02882-t001:** UHPLC–Q/Orbitrap/MS data for compounds from various extracts of *L. multiflora*. Qualitative identification of compounds in *L. multiflora* extracts (R, roots; S, stem bark; DL, dichloromethane leaf; ML, methanol leaf; √, present; X, absent) by UHPLC–Q/Orbitrap/MS analysis.

Compound No.	Rt (min)	Theoretical Mass [M − H]^−^ (*m*/*z*)	Observed Mass [M − H]^−^ (*m*/*z*)	Molecular Formula	MS/MS Fragment Ions (*m*/*z*)	Compound Name	Compound Class	R	S	DL	ML	References
**1**	1.50	483.31	483.12	C_30_H_44_O_5_	483	Poricoic acid B	Triterpenoid	√	√	√	√	[42]
**2**	1.45	455.35	455.30	C_30_H_48_O_3_	454, 453	Boswellic acid alpha	Pentacyclic terpenoid	√	√	√	√	[43]
**3**	10.06	469.36	469.11	C_31_H_50_O_3_	469	Eburicoic acid	Triterpenoid	√	X	X	X	[44]
**4**	1.55	501.32	501.11	C_30_H_46_O_6_	485, 421	Medicagenic acid	Triterpenoids	X	√	X	X	[45]
**5**	1.98	525.16	525.30	C_23_H_28_O_11_	525	Albiflorin	Monoterpenoid glycoside	X	√	X	X	[46]
**6**	0.60	289.07	289.18	C_15_H_14_O_6_	259, 231	Catechin	Flavonoid	X	√	√	X	[47]
**7**	0.60	289.10	289.18	C_15_H_14_O_6_	259, 243	Epicatechin	Flavonoid	X	√	√	X	[48]
**8**	2.12	267.02	267.07	C_22_H_22_O_9_	267	Formononetin-7-*O*-glucoside (Ononin)	Isoflavone glycoside	√	√	√	√	[49]
**9**	2.88	297.23	297.24	C_17_H_14_O_5_	311	3′,4′-Dimethoxy-7-hydroxyflavone	Flavonoid	X	X	X	√	MassBank-BS-BS003750
**10**	1.97	447.09	447.09	C_21_H_20_O_11_	455, 303	Quercitrin	Flavonoid	X	X	√	√	[50]
**11**	1.46	293.17	293.18	C_17_H_26_O_4_	248, 237, 209	6-Gingerol	Phenolic compound	X	X	X	√	[48]
**12**	2.33	277.18	277.20	C_17_H_26_O_3_	283, 233, 205	6-Paradol	Phenolic ketone	√	√	√	√	[48]
**13**	2.58	295.17	295.23	C_16_H_25_NO_4_	-	Esmolol	Phenolic compound	X	X	X	√	[48]
**14**	1.51	305.06	305.92	C_15_H_14_O_7_	289, 221	(−)-Epigallocatechin	Polyphenol	√	√	√	√	[51]
**15**	0.65	427.18	427.18	C_23_H_28_N_2_O_6_	397, 381, 369, 243, 231	Isomajdine	Indole Alkaloid	X	X	√	X	[52]
**16**	0.23	339.17	339.12	C_20_H_24_N_2_O_3_	321, 297	Yohimbic acid	Monoterpene indole alkaloids	X	X	X	√	[53]
**17**	1.46	399.22	339.20	C_22_H_28_N_2_O_5_	325, 309, 304	Reserpic acid	Monoterpene indole alkaloids	√	√	√	√	[53]
**18**	1.52	339.06	339.07	C_15_H_16_O_9_	338	6,7-Dihydroxycoumarin-6-glucoside (Esculin)	Coumarin glucoside	X	X	√	√	[54]
**19**	1.83	307.26	307.19	C_20_H_36_O_2_	289, 290	Eicosadieneoic acid	Polyunsaturated fatty acid	√	√	X	X	[55]
**20**	0.35	229.14	229.14	C_12_H_22_O_4_	213	Dodecanedioic acid	Organic acid	√	√	√	√	[56]

**Table 2 molecules-30-02882-t002:** Comparison of analytical equipment used for compound identification in this study (exp) with the existing literature (lit).

Compound No.	Identified Compound	Molecular Formula	Compound Class	Plant Part(s) (exp)	Plant Part(s) (lit)	Method of Identification (exp)	Method of Identification (lit)	References
**1**	Poricoic acid B	C_30_H_44_O_5_	Triterpenoid	Root, stem bark, leaves	Stem bark, poriae cutis	UHPLC–Q/Orbitrap/MS	UPLC-MS XEVO G2-XS QTOF, UHPLC-QTOF-MS/MS, ^1^H-NMR, ^13^C-NMR	[42,57]
**2**	Boswellic acid alpha	C_30_H_48_O_3_	Pentacyclic terpenoid	Root, stem bark, leaves	Resins	UHPLC–Q/Orbitrap/MS	HPLC-DAD-ESI-MS/MS, HPLC	[43]
**3**	Eburicoic acid	C_31_H_50_O_3_	Triterpenoid	Root	Cactus, cladodes, callus, stem decay	UHPLC–Q/Orbitrap/MS	UPLC-QTOF-MS, FTIR, ^1^H-NMR, ^13^C-NMR	[44,58]
**4**	Medicagenic acid	C_30_H_46_O_6_	Triterpenoids	Stem bark	Root, leaves	UHPLC–Q/Orbitrap/MS	LC ESI FT-ICR MS, FTIR, ^1^H-NMR, ^13^C-NMR	[45,59]
**5**	Albiflorin	C_23_H_28_O_11_	Monoterpenoid glycoside	Stem bark	Whole plant, roots, stems, leaves, flowers	UHPLC–Q/Orbitrap/MS	LC-Q-TOF-MS, LC-IT-MS, HPLC–DAD, HPLC–DAD–ESI–MS	[46,60]
**6**	Catechin	C_15_H_14_O_6_	Flavonoid	Stem bark, leaves	Whole plant	UHPLC–Q/Orbitrap/MS	UHPLC-Q-Orbitrap HRMS, UPLC, LC-MS/MS	[47,61]
**7**	Epicatechin	C_15_H_14_O_6_	Flavonoid	Stem bark, leaves	Pulp, seed, leaves	UHPLC–Q/Orbitrap/MS	LC-ESI-MS/MS,	[48]
**8**	Formononetin-7-*O*-glucoside (Ononin)	C_22_H_22_O_9_	Isoflavone glycoside	Root, stem bark, leaves	Leaves, flowers	UHPLC–Q/Orbitrap/MS	LC-ESI-MS, HPLC/DAD, RP-LC/PDA-ESI-QTOF-MS/MS	[49,62]
**9**	3′,4′-Dimethoxy-7-hydroxyflavone	C_17_H_14_O_5_	Flavonoid	Leaves	N.R.	UHPLC–Q/Orbitrap/MS	LC-ESI-QTOF	MassBank-BS-BS003750
**10**	Quercitrin	C_21_H_20_O_11_	Flavonoid	Leaves	Root barks, trunk barks, stem, leaves, fruits	UHPLC–Q/Orbitrap/MS	LC-MS/MS, UHPLC-ESI-QTOF-MS/MS	[50,63]
**11**	6-Gingerol	C_17_H_26_O_4_	Phenolic compound	Leaves	Pulp, seed, leaves	UHPLC–Q/Orbitrap/MS	LC-ESI-MS/MS, LC-MS/MS Q-TOF	[48]
**12**	6-Paradol	C_17_H_26_O_3_	Phenolic ketone	Root, stem bark, leaves	Pulp, seed, leaves	UHPLC–Q/Orbitrap/MS	LC-ESI-MS/MS, LC-MS/MS Q-TOF	[48]
**13**	Esmolol	C_16_H_25_NO_4_	Phenolic compound	Leaves	Pulp, seed, leaves	UHPLC–Q/Orbitrap/MS	LC-ESI-MS/MS, LC-MS/MS Q-TOF	[48]
**14**	(−)-Epigallocatechin	C_15_H_14_O_7_	Polyphenol	Root, stem bark, leaves	Rhizome bark	UHPLC–Q/Orbitrap/MS	HPLC-MS, SEC-HPLC-UV, HPTLC	[64]
**15**	Isomajdine	C_23_H_28_N_2_O_6_	Indole Alkaloid	Leaves	Root	UHPLC–Q/Orbitrap/MS	NMR	[65]
**16**	Yohimbic acid	C_20_H_24_N_2_O_3_	Monoterpene indole alkaloids	Leaves	Root	UHPLC–Q/Orbitrap/MS	HPLC–ESI–QTOF–MS/MS, UHPLC–UV–MS/MS	[53,66]
**17**	Reserpic acid	C_22_H_28_N_2_O_5_	Monoterpene indole alkaloids	Root, stem bark, leaves	Root	UHPLC–Q/Orbitrap/MS	HPLC–ESI–QTOF–MS/MS	[53]
**18**	6,7-Dihydroxycoumarin-6-glucoside (Esculin)	C_15_H_16_O_9_	Coumarin glucoside	Leaves	Root, young stem, leaves	UHPLC–Q/Orbitrap/MS	HR-LCMS	[54]
**19**	Eicosadieneoic acid	C_20_H_36_O_2_	Polyunsaturated fatty acid	Root, stem bark	Seeds	UHPLC–Q/Orbitrap/MS	LC-Q-TOF-MS	[55]
**20**	Dodecanedioic acid	C_12_H_22_O_4_	Organic acid	Root, stem bark, leaves	Rhizomes	UHPLC–Q/Orbitrap/MS	UHPLC–Q-exactive orbitrap MS	[56]

**Table 3 molecules-30-02882-t003:** The PCA loading scores.

Compounds	PC1	PC2	PC3
**C1**	0	0	0
**C2**	0	0	0
**C3**	−0.279210	−0.73822	−0.61406
**C4**	−0.67912	0.10398	0.72662
**C5**	−0.67912	0.10398	0.72662
**C6**	−0.52758	0.82961	0.18275
**C7**	−0.52758	0.082961	0.18275
**C8**	0	0	0
**C9**	0.88841	−0.21973	0.40304
**C10**	0.82994	0.54927	−0.097482
**C11**	0.88841	−0.21973	0.40304
**C12**	0	0	0
**C13**	0.88841	−0.21973	0.40304
**C14**	0	0	0
**C15**	0.069923	0.85397	−0.5156
**C16**	0.88841	−0.021973	0.40304
**C17**	0	0	0
**C18**	0.82994	0.54927	−0.097482
**C19**	−0.82994	−0.54927	0.097482
**C20**	0	0	0

**Table 4 molecules-30-02882-t004:** Antioxidant activity of *Lippia multiflora* extracts.

Samples	DPPH IC_50_ (µg/mL)	Reducing Power IC_0.5_ (µg/mL)
Root extract	25.108 ± 19.798 ^ab^	0.218 ± 0.157 ^a^
Stem bark extract	5.031 ± 2.940 ^a^	0.029 ± 0.026 ^a^
Methanol leaf extract	0.559 ± 0.269 ^a^	0.710 ± 1.220 ^a^
Dichloromethane leaf extract	6.299 ± 0.495 ^a^	0.326 ± 0.210 ^a^
Ascorbic acid	44.308 ± 9.813 ^b^	0.355 ± 0.393 ^a^
Gallic acid	37.340 ± 17.529 ^ab^	8.210 ± 11.257 ^a^
Quercetin	37.361 ± 21.356 ^b^	3.760 ± 2.979 ^a^

Notes: A different superscript letter indicates significant differences using a one-way ANOVA at *p* < 0.05. Data (n = 3) expressed as mean ± standard deviation. For DPPH (2,2-diphenyl-1-picrylhydrazyl) free radical scavenging activity: ^a^—stem bark crude extract was not significantly different from the methanol leaf extract and dichloromethane leaf extract; ^b^—ascorbic acid was not significantly different from quercetin; ^ab^—root extract was not significantly different from all crude extracts, ascorbic acid, gallic acid, and quercetin. For reducing power activity: ^a^—root extract was not significantly different from all crude extracts, ascorbic acid, gallic acid, and quercetin.

**Table 5 molecules-30-02882-t005:** Comparison of the methods used to determine the antioxidant activity of extracts obtained in this study (exp) with the existing literature (lit).

Compound No.	Identified Compound	Molecular Formula	Compound Class	Plant Part(s) (exp)	Plant Part(s) (lit)	Antioxidant Assay (exp)	Antioxidant Assay (lit)	References
1	Poricoic acid B	C_30_H_44_O_5_	Triterpenoid	Root, stem bark, leaves	Stem bark	DPPH, RPA	DPPH	[42]
**2**	Boswellic acid alpha	C_30_H_48_O_3_	Pentacyclic terpenoid	Root, stem bark, leaves	Resins	DPPH, RPA	DPPH, ABTS	[74]
**3**	Eburicoic acid	C_31_H_50_O_3_	Triterpenoid	Root	-	DPPH, RPA	-	-
**4**	Medicagenic acid	C_30_H_46_O_6_	Triterpenoids	Stem bark	-	DPPH, RPA	-	-
**5**	Albiflorin	C_23_H_28_O_11_	Monoterpenoid glycoside	Stem bark	Root, stem, leaves, flowers	DPPH, RPA	DPPH, ABTS	[60]
**6**	Catechin	C_15_H_14_O_6_	Flavonoid	Stem bark, leaves	N.R.	DPPH, RPA	ABTS, FRAP	[68]
**7**	Epicatechin	C_15_H_14_O_6_	Flavonoid	Stem bark, leaves	N.R.	DPPH, RPA	ABTS, FRAP	[68]
**8**	Formononetin-7-*O*-glucoside (Ononin)	C_22_H_22_O_9_	Isoflavone glycoside	Root, stem bark, leaves	Leaves, flowers	DPPH, RPA	DPPH, ABTS	[62]
**9**	3′,4′-Dimethoxy-7-hydroxyflavone	C_17_H_14_O_5_	Flavonoid	Leaves	-	DPPH, RPA	-	-
**10**	Quercitrin	C_21_H_20_O_11_	Flavonoid	Leaves	Trunk barks	DPPH, RPA	ABTS, ORAC, TEAC	[50,69,70]
**11**	6-Gingerol	C_17_H_26_O_4_	Phenolic compound	Leaves	Rhizomes	DPPH, RPA	DPPH	[67]
**12**	6-Paradol	C_17_H_26_O_3_	Phenolic ketone	Root, stem bark, leaves	-	DPPH, RPA	-	-
**13**	Esmolol	C_16_H_25_NO_4_	Phenolic compound	Leaves	-	DPPH, RPA	-	-
**14**	(−)-Epigallocatechin	C_15_H_14_O_7_	Polyphenol	Root, stem bark, leaves	Rhizome bark	DPPH, RPA	DPPH	[64]
**15**	Isomajdine	C_23_H_28_N_2_O_6_	Indole Alkaloid	Leaves	Root	DPPH, RPA	ABTS, DMPD, Fe^2+^	[65]
**16**	Yohimbic acid	C_20_H_24_N_2_O_3_	Monoterpene indole alkaloids	Leaves	-	DPPH, RPA	-	-
**17**	Reserpic acid	C_22_H_28_N_2_O_5_	Monoterpene indole alkaloids	Root, stem bark, leaves	-	DPPH, RPA	-	-
**18**	6,7-Dihydroxycoumarin-6-glucoside (Esculin)	C_15_H_16_O_9_	Coumarin glucoside	Leaves	-	DPPH, RPA	-	-
**19**	Eicosadieneoic acid	C_20_H_36_O_2_	Polyunsaturated fatty acid	Root, stem bark,	-	DPPH, RPA	-	-
**20**	Dodecanedioic acid	C_12_H_22_O_4_	Organic acid	Root, stem bark, leaves	-	DPPH, RPA	-	-

Notes: N.R.—not reported; DPPH—DPPH (2,2-diphenyl-1-picrylhydrazyl) free radical scavenging assay; RPA—reducing power assay; ABTS—2,2′-azino-bis(3-ethylbenzothiazoline-6 sulfonic acid) assay; FRAP—ferric reducing antioxidant power; DMPD—N,N dimethyl-p-phenylenediamine radical scavenging; Fe^2+^—ferrous ions chelating activity; ORAC—oxygen radical absorbance capacity; and TEAC—Trolox equivalent antioxidant capacity.

**Table 6 molecules-30-02882-t006:** Pearson’s correlation of phytochemicals (TFC and TPC) and their antioxidant activities.

Test	TFC	TPC	DPPH	RP
TFC	1			
TPC	−0.111 *	1		
DPPH	0.619 *	0.071 *	1	
RP	−0.956	−0.019	−0.406	1

Notes: Correlation coefficient (r) between the phytochemicals (TFC and TPC) with antioxidant capacity (DPPH and RP). If the r value is close to +1 or −1, it indicates a strong positive relationship. If r is near 0, it indicates a weak or no relationship. * Correlation is significant at *p* < 0.05. Abbreviations: TFC—total flavonoid content; TPC—total phenolic content; DPPH—DPPH (2,2-diphenyl-1-picrylhydrazyl) free radical scavenging assay; and RP—reducing power capacity.

**Table 7 molecules-30-02882-t007:** Inhibitory concentrations (IC_50_ in µg/mL) of extracts of the root, stem bark, and leaf of *Lippia multiflora* and doxorubicin (positive control) in Vero monkey kidney cell lines.

Samples	Vero IC_50_ (µg/mL)
Root extract	37.15 ± 2.61 ^b^
Stem bark extract	273.60 ± 1.70 ^e^
Methanol leaf extract	74.22 ± 2.33 ^c^
Dichloromethane leaf extract	132.90 ± 4.52 ^d^
Doxorubicin (positive control)	4.97 ± 0.83 ^a^

Notes: The values above are presented by mean ± standard deviation of three replicates (n = 3). Values with different superscript letters in each column are significantly different using one-way ANOVA at *p* < 0.05.

## Data Availability

Data are contained within the article and Appendix A.

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
