# Peer review of "Phytochemical Screening and Biological Activities of Lippia multiflora Moldenke"

_molecules, 2025, doi:10.3390/molecules30132882_

Round 1
Reviewer 1 Report
Comments and Suggestions for Authors
1. Keywords should be narrowed down to five.
2. The background section should be described in paragraphs; all the paragraphs in the full text are too long.
3, All the diagrams should be made clearer.
4, Cytotoxicity was tested with only one type of cell, which is not enough proof power, and liver cells should be added.
5, Figures 2-5 can be combined into one figure.
6. Comparison with existing studies should be added, such as the ingredients found, the efficacy of the
7, In the cytotoxicity test, the unit used for the positive agent is um, which is inconsistent with the unit of the plant extract (ug/ml). This does not provide a visual measure of the toxicity intensity of the plant extract.
Author Response
"Please see the attachment."

Reviewer 2 Report
Comments and Suggestions for Authors
After reviewing the manuscript "Phytochemical Screening and Biological activities of Lippia multiflora Moldenke", I have the following comments:
- The study investigates the phytochemical composition, antioxidant activity, and cytotoxicity of root, stem bark, and leaf extracts of Lippia multiflora using UHPLC–Q/Orbitrap/MS, identifying 20 bioactive compounds. It highlights the plant’s potential as a natural source of antioxidants with moderate cytotoxicity at higher concentrations.
- The paper repeatedly claims “first-time” findings (e.g., first report of total flavonoid content in stem/root; first UHPLC profiling), which are not always well-supported. Some compounds (like catechin, quercetin, etc.) are widely reported in similar species and even L. multiflora literature verification is insufficient.
- Several comparisons claim statistical significance but do not include p-values or confidence intervals.
- While the study reports antioxidant and cytotoxic activities, there is no proper correlation analysis (e.g., regression or PCA loadings) connecting specific compounds with observed activities.
- Supplementary figures (S1–S24) are referred to but not critically discussed or well-labeled in the text. Mass spectra are not annotated with molecular fragments, making verification impossible for readers.
- While previous studies are cited, comparative tables summarizing differences in compound identification or antioxidant activity with past works would strengthen the manuscript.
- Some references used to justify findings (e.g., [60], [61]) are weakly connected and often involving different plant systems, making comparisons with L. multiflora tenuous.
- Figure 2:The PCA-biplot is missing loading vectors that indicate which variables (metabolites) are most influential along PC1 and PC2 axes. Without this, the variance interpretation is incomplete.
- The clustering pattern shown in Figure 2 is not sufficiently interpreted in the text. It mentions correlation but does not discuss biological significance or which metabolites drive the separation.
Author Response
"Please see the attachment."

Round 2
Reviewer 1 Report
Comments and Suggestions for Authors
The authors have revised the manuscript in response to comments. It is recommended that it be accepted
Author Response
"Please see the attachment."

Reviewer 2 Report
Comments and Suggestions for Authors
After reviewing the revised manuscript, I have the following comments:
- The authors have made substantial revisions and addressed most major criticisms. However, several issues remain.
-
Although supplementary figures are referenced and labeled, they still lack detailed annotation of fragment ions or molecular structures, which impairs verification and transparency. For example: MS spectra like Figure S5 (Poricoic acid B) and S10 (Catechin) show peaks but lack annotated fragment assignments.
- While PCA and correlation plots are added, no regression models or quantitative analysis linking individual compounds to antioxidant/cytotoxic activities is provided. Also, the PCA biplot is incomplete without loading vectors (arrows) indicating which compounds (variables) contribute most to PC1 and PC2. Additionally, no confidence intervals or ellipses are shown around clusters, which are typically included to indicate variability or statistical significance of groupings.
- The PCA and correlation plot methods are insufficiently explained: type of scaling used, software specifics, number of replicates per extract not stated.
-
Despite the removal of “first-time” claims in the main text, the abstract and introduction still imply novelty: Example: “UHPLC–Q/Orbitrap/MS compositions of L. multiflora have also not been explored.” This may be misleading since prior work does exist.
Author Response
"Please see the attachment."
